# Application value of volumetric CT value in quantifying the activity of a pulmonary tuberculoma

**Ganhui Wei** *, **Jiacheng Zhang, Xiaowei Qiu**

Hang zhou Red Cross Hospital, Hangzhou City, Zhejiang Province, China

* weiganhui@163.com

## Abstract

### Objective

The purpose of this study was to explore the auxiliary diagnostic value of volumetric CT value in quantifying the activity of a pulmonary tuberculoma.

### Methods

Chest CT image data of 112 patients with pulmonary tuberculomas who were diagnosed clinically between October 16, 2013 and March 21, 2023 were selected. With the shortest diameter axis>5 mm on the mediastinal window serving as the inclusion criterion, 108 active tuberculomas and 64 non-active tuberculomas were selected. The focused image was manually segmented using ITK-SNAP software, the volumetric CT value of the focus was calculated, and the ROC curve was analyzed. Using the final clinical diagnosis as the reference standard, the auxiliary diagnostic efficacy and consistency of the conventional CT film reading method and volumetric CT value in determining the activity of a pulmonary tuberculoma were compared.

### Results

The volumetric CT value of 108 active pulmonary tuberculoma lesions (33.39 [28.17,36.23] HU) was significantly less than 64 inactive pulmonary tuberculoma lesions (78.91 [57.81,120.31] HU); the difference was statistically significant (Z = −10.888. P < 0.001). ROC curve analysis showed that at a maximum Yoden index value of 0.963, the optimal volumetric CT threshold value was 45.32 HU, the sensitivity and specificity of the volumetric CT value in determining the activity of a pulmonary tuberculoma were 97.2% and 100.0%, respectively, and the maximum area under the ROC curve was 0.998. Taking the final clinical diagnosis as the reference standard, the sensitivity, specificity, consistency, and kappa value of the conventional CT film reading method for determining the activity of a pulmonary tuberculoma were 72.2% (78/108), 70.3% (45/64), 71.5% (123/172), and 0.413, respectively, while the corresponding volumetric CT values were 97.2% (105/108), 100.0% (64/64), 98.3% (168/172), and 0.951, respectively.

**Data Availability Statement:** All relevant data are within the manuscript and its Supporting Information files.

**Funding:** This research received funding from the Zhejiang Provincial Health Science and Technology Plan Project (2021KY252). The funders had no role in study design, data collection and analysis, decision to publish or preparation of the manuscript.

**Competing interests:** No authors have competing interests.

## Conclusion

Accurately quantifying the volumetric CT value of a pulmonary tuberculoma focus determines the activity of a pulmonary tuberculoma, which has very important auxiliary diagnostic value.

## Introduction

Tuberculosis is one of the top ten lethal medical conditions worldwide. According to the 2022 World Health Organization Global Tuberculosis Report [1], there were approximately 10.6 million new tuberculosis patients worldwide in 2021, which was an increase of 4.5% compared to 2020. The number of deaths was approximately 1.6 million, and the mortality rate was as high as 15% [1]. Therefore, the early diagnosis of pulmonary tuberculosis activity and early treatment is the key to controlling tuberculosis. At present, CT scanning remains the preferred modality with which to determine the activity of pulmonary tuberculosis in clinical practice; however, because the conventional CT film reading method mainly determines the activity of pulmonary tuberculosis by morphology, there is no quantitative standard and the need for clinical precise treatment cannot be satisfied. Specifically, conventional CT cannot accurately determine the activity of pulmonary tuberculomas [2], which severely impairs the physical and mental health of patients with a pulmonary tuberculoma. Tuberculoma is a well-defined nodule or mass in the lung caused by *Mycobacterium tuberculosis*. Pathologically, a pulmonary tuberculoma is a case-like mass wrapped by multiple concentric layers of connective tissue with no encompassing inflammation [3]. In routine CT diagnosis, the diagnosis of pulmonary tuberculosis activity is divided into active, inactive, and uncertain active. Tuberculoma belongs to the uncertain active diagnosis category [4]. Currently, clinicians mainly assess the activity of a pulmonary tuberculoma based on chest CT findings combined with clinical symptoms, blood test results, and re-examination after anti-tuberculosis treatment. Therefore, quantification of chest volumetric CT values of patients with pulmonary tuberculoma was implemented by the author to determine whether this method has auxiliary diagnostic value for determining the activity of pulmonary tuberculomas and whether the volumetric CT value method satisfies precision medical treatment requirements.

## Materials and methods

### Research object

The chest CT image data from 112 patients with pulmonary tuberculoma who were diagnosed clinically at Hang zhou Red Cross Hospital between October 16, 2013 and March 21, 2023were analyzed retrospectively. All patients were enrolled from the Tuberculosis Outpatient or Inpatient Departments of Hangzhou Red Cross Hospital. The selected tuberculoma lesions in this study include tuberculomas without cavities and tuberculomas with thick walled cavities (cavity wall thickness >5mm), excluding fibroproliferative lesions, exudative lesions, thin-walled cavities (cavity wall thickness <5mm), and miliary pulmonary tuberculosis. All patients in this study had follow-up evaluations. Data analysis was conducted in April 2023. The regions of interest (ROIs) of tuberculomas were delineated using ITK-SNAP software (www.itksnap.Org). During the 2-year follow-up period, 60 patients with active pulmonary tuberculomas (total = 108 active pulmonary tuberculomas) were identified, including 31 males (51.7%) and 29 females (48.3%) with a median age of 33(26,40) years. There were 52 patients with inactive pulmonary tuberculomas (total = 64 inactive pulmonary tuberculomas),

including 28 males (53.8%) and 24 females (46.2%) with a median age of 33 (31,38) years. There was no statistically significant difference in gender and age between the two groups (x = 0.106, P = 0.745; Z = −0.922, P = 0.357). This study was approved by the Medical Ethics Committee of Hangzhou Red Cross Hospital. The data were anonymous, and the requirement for informed consent was therefore waived.

The inclusion criteria were as follows: (1) All pulmonary tuberculomas were diagnosed by sputum smear, sputum GeneXpert MTB/RIF detection, histopathology or CT scanning, and effective anti-tuberculosis drug treatment. (2) The included tuberculomas had clear boundaries, and the shortest diameter line on the mediastinal window was > 5mm, but it was considered difficult to assess pulmonary tuberculoma activity based on the conventional CT film reading method. (3) All imaging data of patients with pulmonary tuberculomas were re-examined by chest CT after standard anti-tuberculosis drug treatment.

The criteria for judging the activity of tuberculomas were as follows: after 2 years of follow-up evaluations, the patients in the group were judged to have an inactive pulmonary tuberculoma if the tuberculoma focus was stable, there were no clinical symptoms and signs related to active pulmonary tuberculosis, and the bacteriologic examination was negative [5]; and if the stability of the tuberculoma cannot be established and enlarged or decreased in size(except for the possibility of secondary diseases), the focus of the tuberculoma was judged to be an active pulmonary tuberculoma regardless of whether there were clinical symptoms and signs related to active pulmonary tuberculosis and whether the bacteriologic examination is positive or negative [5].

## Research methods

1) Image acquisition: The CT machine used for inspection was a GE Bright Speed Elite 16-row multi-slice spiral CT (GE Healthcare, Milwaukee, WI, USA). The tube voltage was 120 kV, the tube current was 150 mAs, and the pitch was 0.81–0.94. The patients were trained to hold their breath at the end of inspiration, and the scanning range was from the tip-to-bottom of the lung. The scanning layer thickness and interval were 2.5 mm. The reconstruction layer thickness of the mediastinal window was 5 mm, and the matrix was $512 \times 512$.

After scanning, the CT image was imported into the ITK-SNAP software in DICOM format to outline the ROIs of tuberculoma lesions. When delineating the ROI, the soft tissue density and calcification density were selected on the mediastinal window for delineation, and 1–2 pixels were retreated in each direction of the lesion boundary to reduce the impact of the partial volume effect. All operations were performed on the CT cross-section without 3D reconstruction. The volumetric CT value of the focus was calculated using the software volume and statistical option, then the measured volumetric CT value of the pulmonary tuberculoma was imported into SPSS25.0 software. The average value option to obtain the average volumetric CT value was selected. The ROI did not include normal lung tissue, cavity, gas, and other components without pathologic tissue to avoid errors. In addition, calcifications were included in the measurement in this study and it was shown that the activity of individual lesions based on volumetric CT values was inconsistent with the clinical follow-up results; however, according to the statistical data, the ROI, including calcifications, did not have a significant impact on the results. The delineation of the tuberculoma focus ROI and the measurement of the tuberculoma focus volumetric CT value were performed by 2 chief radiologists with >10 years of experience in the diagnosis of tuberculosis and 1 deputy chief radiologist with >20 years of experience in the diagnosis of tuberculosis in strict accordance with the above standards. If there were differences on the boundary, the differences were discussed among the radiologists and the judgment of the deputy chief radiologist prevailed.

2) Image analysis: All CT images were read by two senior deputy chief radiologists in a double-blind fashion to analyze the CT image signs and determine the pulmonary tuberculoma activity. If the conclusions were not consistent, the Radiology Department reached an agreement through collective discussion. The correct rate = the number of tuberculoma lesions correctly judged/the number of all tuberculoma lesions ×100%.

## Statistical processing

SPSS25.0 software was used for statistical analysis of the data. The measurement data of a non-normal distribution is described by the median (quartile), and the differences between groups were compared using the Mann-Whitney U test with a $P < 0.05$ as the difference with statistical significance. The receiver operating characteristic (ROC) curve was used to predict the optimal classification threshold of pulmonary tuberculoma activity, and the sensitivity, specificity, and area under the ROC curve (AUC) were calculated, as well as the detection efficiency and consistency kappa test of the two methods. Akappa value ≥0.75 indicated good consistency, 0.4∼0.75 indicated general consistency, and <0.4 indicated poor consistency.

## Results

### Volumetric CT value detection of active and inactive tuberculomas

The average volumetric CT value of 108 active tuberculomas (33.39 [28.17,36.23] HU) was significantly lower than the average volumetric CT value of 64 inactive tuberculomas (78.91 [57.81,120.31] HU);the difference was statistically significant (Z = −10.888, P < 0.001).

### ROC curve analysis of volumetric CT value of active and inactive tuberculomas

The volumetric CT value of active and inactive pulmonary tuberculomas was further analyzed by drawing ROC curves. The maximum Yoden index value (sensitivity + specificity− 1) was 0.963, the optimal critical value was 45.32 HU, the diagnostic sensitivity for the determination of pulmonary tuberculoma activity was 97.2%, the specificity was 100.0%, and the maximum area under the ROC curve was 0.998 (Fig 1).

### Determining pulmonary tuberculoma activity based on routine CT film reading and the volumetric CT value

According to the diagnostic criteria of focus activity, 108 active and 64 inactive tuberculomas were among the 172 pulmonary tuberculomas. There were 64 pulmonary tuberculomas with calcifications and 24 pulmonary tuberculomas with thick-walled cavities. A total of 123 tuberculomas were correctly diagnosed using the conventional CT film reading method, with a correct rate of 71.5%. Among the 123 tuberculomas, 78 were active pulmonary tuberculomas and 45 were inactive pulmonary tuberculomas. The sensitivity, specificity, and consistency were 72.2%, 70.3%, and 71.5%, respectively. The correct number of tuberculomas based on volumetric CT value was 168 (97.7%). Among the 168 tuberculomas, there were 104 active tuberculomas and 64 inactive tuberculomas;3 lesions that were misjudged were active tuberculoma with calcifications. The sensitivity, specificity, and consistency rates were 97.2%, 100.0%, and 98.3%, respectively (Table 1).

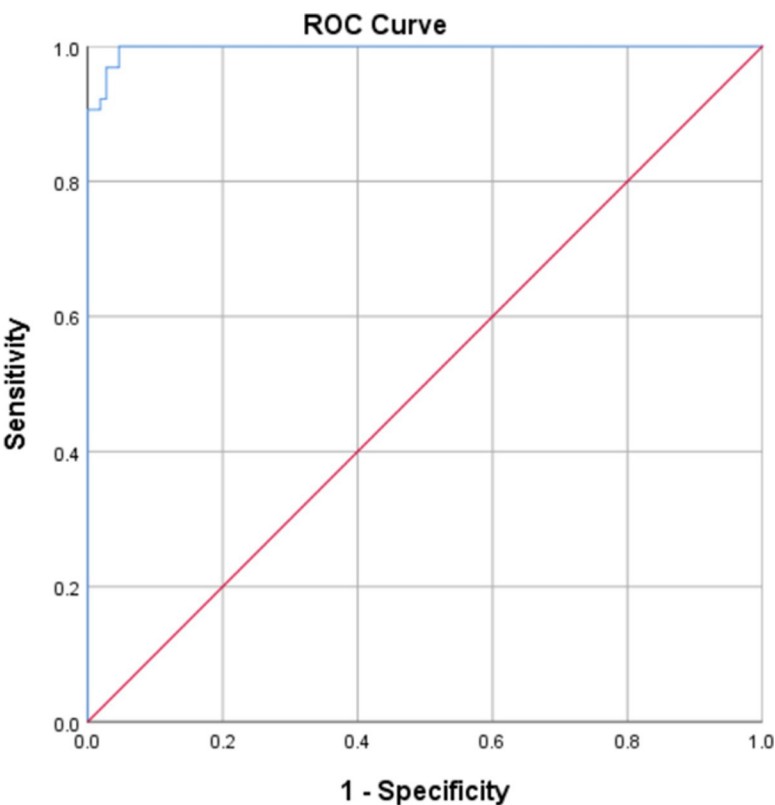

**Fig 1. ROC curve of volumetric CT value in the judgment of pulmonary tuberculoma activity.**

## Clinical examples of two methods to determine pulmonary tuberculoma activity

In cases 1–4 chest CT scanning was performed, the activity was determined using the conventional CT film reading method, and all cases were misdiagnosed. The determination of activity

**Table 1. Diagnostic efficacy of determining pulmonary tuberculoma activity with the final diagnosis result as a reference for the conventional CT film reading method and volumetric CT value.**

| Method | Tuberculoma (unit) | | Sensitivity (%) | Specificity (%) | PPV (%) | NPV (%) | Consistency (%) | *Kappa* value |
|---|---|---|---|---|---|---|---|---|
| | Active | Inactive | | | | | | |
| Conventional method | 108 | 64 | 72.2 | 70.3 | 80.7 | 59.9 | 71.5 | 0.413 |
| Active | 78 | 19 | | | | | | |
| Inactive | 30 | 45 | | | | | | |
| Volumetric CT value | 108 | 64 | 97.2 | 100.0 | 100.0 | 94.1 | 98.3 | 0.951 |
| Active | 105 | 0 | | | | | | |
| Inactive | 3 | 64 | | | | | | |

Note: Sensitivity = number of true positives/(number of true positives + number of false negatives) × 100%;Specificity = number of true negatives/(number of true negatives + number of false positives) × 100%; Positive predictive value = number of true positives/(number of true positives + number of false positives) × 100%; Negative predictive value = true negative number/(true negative number + false negative number) × 100%; Consistency rate = (number of true positives + number of true negatives)/total number × 100%.

Positive predictive value (PPV).

Negative predictive value (NPV).

with 45.32 HU as the critical value of volumetric CT was consistent with the final clinical diagnosis of the lesion change after 2 years of follow-up evaluation, and the diagnostic accuracy was significantly improved (Figs 2–18).

Figs 2–4. A 39-year-old male patient was misdiagnosed with inactive upper left pulmonary tuberculoma based on conventional CT scanning. Fig 2 shows a nodule in the left upper lung on chest CT which has a clear boundary and calcifications can be seen in the lesion. Fig 3 shows the ROI along the edge of the lesion and filled with blue pixels (arrow). Fig 4 shows that the volumetric CT value of blue pixel filled lesions calculated by ITK-SNAP software is 44.49 HU (<45.32 HU was interpreted as active pulmonary tuberculoma, and the follow-up evaluation showed significant absorption of the lesions and the result was consistent).

Figs 5–7. A 41-year-old male patient was misdiagnosed with active lower right pulmonary tuberculoma based on conventional CT scanning. Fig 5 shows a small nodule in the right lower lung on chest CT with no obvious calcification in the lesion. Fig 6 shows the ROI along the edge of the lesion and filled with blue pixels (arrow). Fig 7 shows that the volumetric CT value of blue pixel filled lesions calculated by ITK-SNAP software is 49.35 HU (>45.32 HU was interpreted as inactive pulmonary tuberculoma, and the follow-up evaluation showed that the lesion was stable with consistent results).

Figs 8–10. A 42-year-old male was misdiagnosed with active upper right pulmonary tuberculoma based on conventional CT scanning. Figs 8 and 9 show a nodule with a thick-walled cavity in the right upper lung on the lung and mediastinal windows of the chest CT. Fig 10 shows the volumetric CT value of the lesion calculated by ITK-SNAP software is 61.42 HU (>45.32 HU was interpreted as inactive pulmonary tuberculoma); the lesion was stable after 2 years of follow-up.

Figs 11–18. A 60-year-old female patient was misdiagnosed as active lower right pulmonary tuberculoma based on conventional CT scanning. Figs 11 and 15 show the lung window and mediastinal window of the patient's chest CT initially diagnosed in 2016 showing the patchy nodular high-density shadow in the dorsal segment of the right lower lung, with clear boundaries and a visible "tree bud sign," as shown by the long arrow. Figs 12–14 and 16–18 show the lung and mediastinal windows of chest CT after 1 week, 1 year and 2 years of treatment,

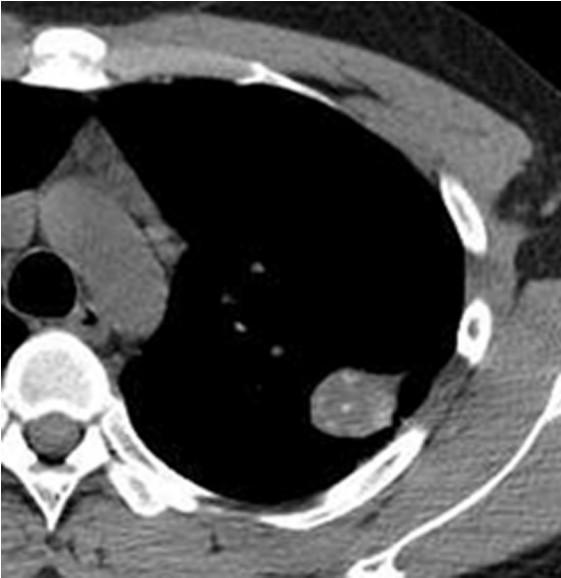

**Fig 2. A tuberculoma in the left upper lung on chest CT.**

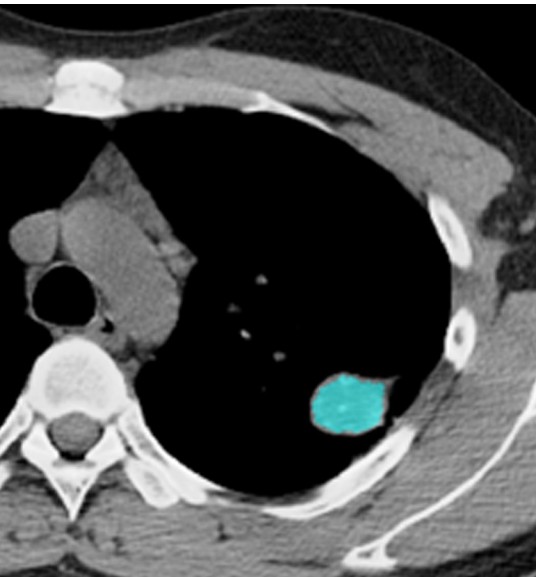

**Fig 3. The ROI along the edge of the lesion and filled with blue pixels.**

respectively. The measured volumetric CT value was >45.32 HU (it was interpreted as inactive pulmonary tuberculoma and the focus was stable during the follow-up evaluation with the same results).

## Discussion

At present, the main basis for clinicians to judge active pulmonary tuberculosis by chest CT is based on the focus with ground-glass changes, the tree bud sign, the central lobular nodule sign, consolidation, and a thick-walled cavity. The main basis for judging inactive pulmonary tuberculosis is based on the focus with fibrosis and calcifications, which has high diagnostic value and is the main method for clinicians to judge the activity of pulmonary tuberculosis [6].

Volumes and Statistics - ITK-SNAP

| | Label Name | Voxel Count | Volume (mm3) | Intensity Mean ± SD (2.5mm std ) |
|---|---|---|---|---|
| 0 | Clear Label | 36436190 | 5.018e+07 | -1016.2496±1126.6900 |
| 1 | Label 1 | 1826 | 2515 | 44.4923±24.4620 |

**Fig 4. The volumetric CT value of blue pixel filled lesions calculated by ITK-SNAP software.**

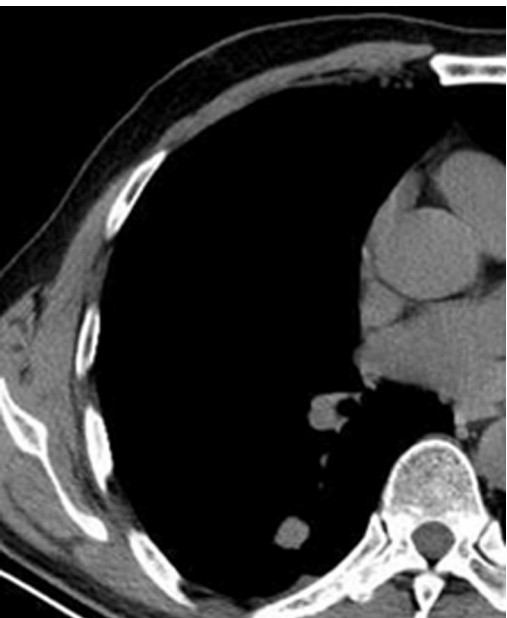

**Fig 5. A small tuberculoma in the right lower lung on chest CT.**

Because this method is mainly used to determine the activity of pulmonary tuberculosis from the shape of the lesion, and there is no quantitative standard, it is often impossible to make an accurate judgment of the activity of a pulmonary tuberculoma with clear boundaries. A PET-CT scan has diagnostic value, but the radiation exposure is significant, PET-CT scans are expensive, and PET-CT scans cannot be routinely used in clinical practice. The activity of tuberculomas can be accurately quantified using the volumetric CT value. The smaller the volumetric CT value, the greater the possibility of an active pulmonary tuberculoma. The larger

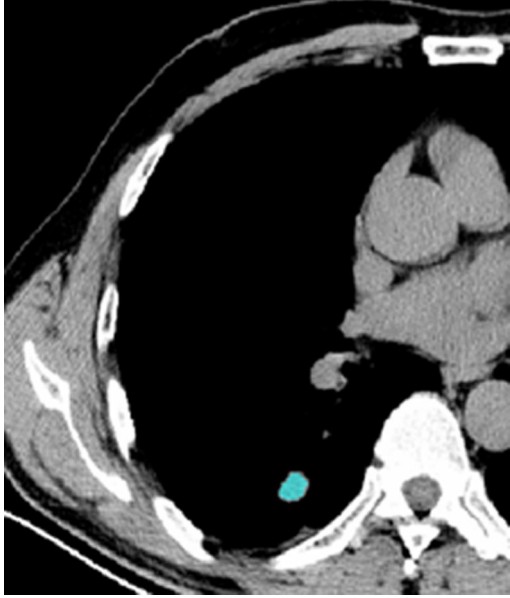

**Fig 6. The ROI along the edge of the lesion and filled with blue pixels.**

Volumes and Statistics - ITK-SNAP

| | Label Name | Voxel Count | Volume (mm3) | Intensity Mean ± SD (2.5mm std ) |
|---|---|---|---|---|
| 0 | ■ Clear Label | 36437336 | 5.018e+07 | -1016.2163±1126.6879 |
| 1 | ■ Label 1 | 680 | 936.4 | 49.3456±35.3762 |

**Fig 7. The volumetric CT value of blue pixel filled lesions calculated by ITK-SNAP software.**

the volumetric CT value, the greater the possibility of an inactive pulmonary tuberculoma. In the current study, the sensitivity, specificity, accuracy, and kappa value of the conventional CT film reading method to determine the activity of pulmonary tuberculomas were 72.2%, 70.3%, 71.5%, and 0.413, respectively. The sensitivity, specificity, accuracy, and kappa value of judging the activity of tuberculomas based on a volumetric CT value were 97.2%, 100.0%, 98.3%, and 0.951, respectively, which were significantly higher than the conventional CT film reading method. The value of determining the activity of tuberculomas based on the volumetric CT was high, especially when the threshold value was 45.32 HU; the sensitivity, specificity, accuracy, and kappa value of judging pulmonary tuberculoma activity was optimal.

Volumetric CT value analysis can be used to determine the activity of a tuberculoma focus, which is determined by the pathologic composition of a pulmonary tuberculoma. The pathologic composition of a tuberculoma is composed of caseous lesions, granulomatous lesions, fibrous lesions, and calcifications [3, 7]. The caseous necrosis lesions contain many lipids [3,

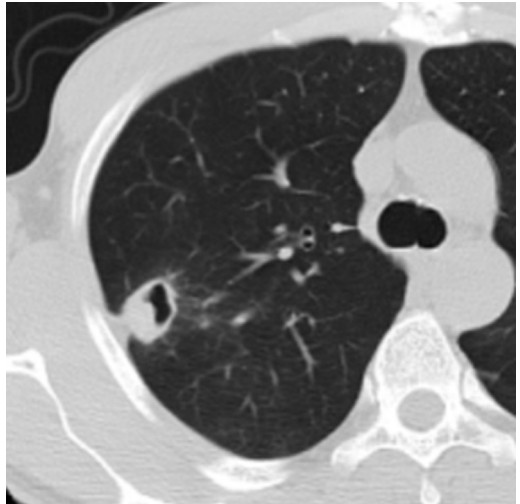

**Fig 8. A tuberculoma with a thick-walled cavity on the lung windows of the chest CT.**

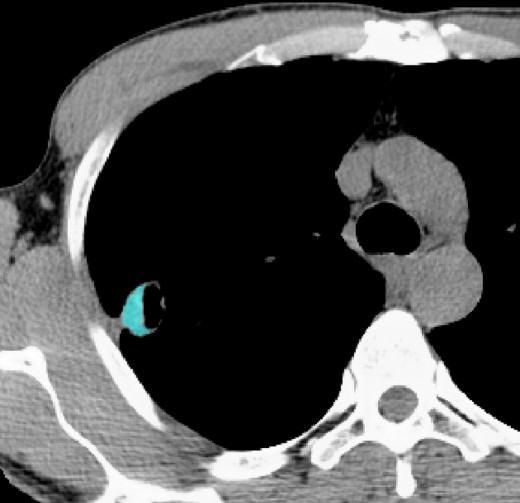

**Fig 9. A tuberculoma with a thick-walled cavity on the mediastinal windows of the chest CT.**

7], and the CT value is low (negative). For granulomatous lesions, the CT value is low (20–40 HU), while for fibrous (40–80 HU) and calcified lesions (>80 HU), the CT value is high [3, 7]. Therefore, an analysis of volumetric CT values accurately distinguished active tuberculomas from inactive tuberculomas. Lee et al. [6] made dynamic observations on high-resolution CT films of 52 sputum-positive pulmonary tuberculosis patients from before or after <2 weeks of antituberculosis treatment to 6–9 months of antituberculosis treatment and reported that tuberculosis foci were loose and low density before anti-tuberculosis treatment. After anti-tuberculosis treatment, with an extension of treatment time, the tuberculosis focus shrinks, the density increases, and the calcification focus often increases, which is consistent with the results of the current study. Sineglazov et al. [8] collected the CT images of 9000 patients with pulmonary tuberculomas for marking, training, and modeling, then judged the activity of tuberculomas through artificial intelligence. Sineglazov et al. [8] found that the density of active tuberculomas was low and the density of inactive pulmonary tuberculomas was high.

Volumes and Statistics - ITK-SNAP

| | Label Name | Voxel Count | Volume (mm3) | Intensity Mean ± SD (2.5mm std) |
|---|---|---|---|---|
| 0 | ■ Clear Label | 36437529 | 5.018e+07 | -1016.2108±1126.6874 |
| 1 | ■ Label 1 | 487 | 670.7 | 61.4230±35.9155 |

**Fig 10. The volumetric CT value of the lesion calculated by ITK-SNAP software.**

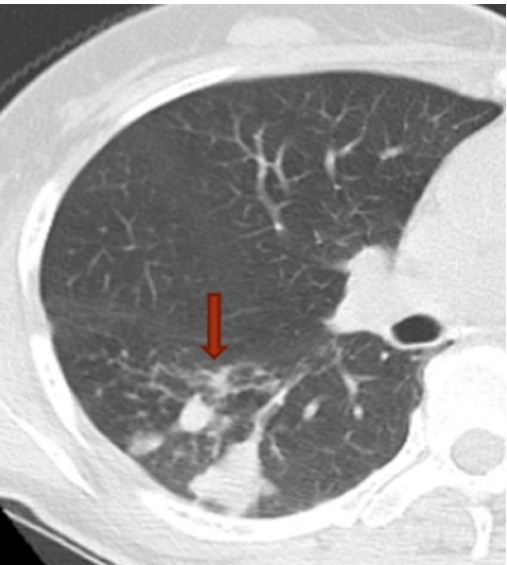

**Fig 11. The tuberculomas in the dorsal segment of the right lower lung.**

Therefore, according to the density of a tuberculoma, that is the CT value, the activity of tuberculomas was divided into three levels (low, medium, and high). The lower the CT value of a tuberculoma, the higher the activity. The higher the CT value, the lower the activity. Finally, the standard was tested in multiple tuberculoma samples and the accuracy of the method was >96%, which is also consistent with the results of this study. In the process of anti-tuberculosis treatment, except for a few tuberculomas that were completely absorbed after anti-tuberculosis treatment, the majority of active tuberculomas transformed from a soft tuberculoma focus to a hard tuberculoma focus [9]. Tuberculomas are absorbed during treatment, the volume decreases, and the density increases [9]. When the density and volumetric CT values reach a specific threshold, the focus will be in a stable state, the patient's symptoms will disappear, the

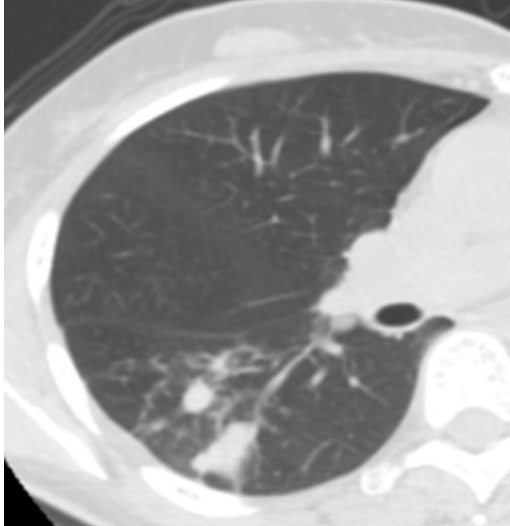

**Fig 12. The tuberculomas on the lung window of chest CT after 1 week of treatment.**

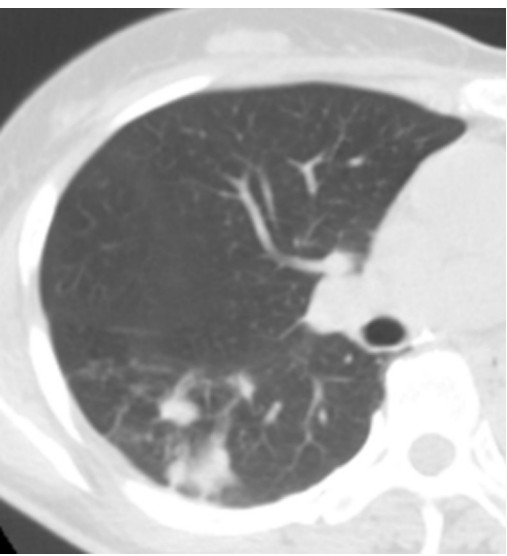

**Fig 13. The tuberculomas on the lung window of chest CT after 1 year of treatment.**

bacteriological examination becomes negative, and the active tuberculoma will become an inactive tuberculoma [9].

On the basis of conventional imaging analysis methods, this study used ITK-SNAP software to introduce an analysis of volumetric CT values, which further improved the imaging diagnostic efficiency of judging the activity of a pulmonary tuberculoma. ITK-SNAP software is an interactive medical image segmentation tool that is mainly used for medical image pre-processing, segmentation, and registration. Li et al. [10] using ITK-SNAP software to segment the ultrasonic images of 150 cases of thyroid cancer confirmed by pathologic evaluation, the imaging characteristics for the diagnosis of thyroid cancer were extracted. The results showed that

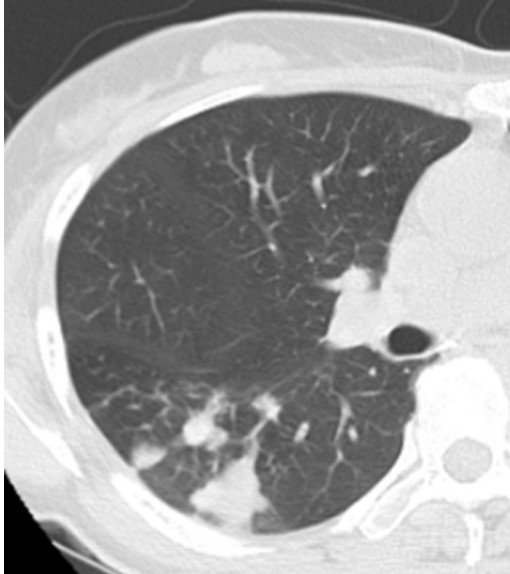

**Fig 14. The tuberculomas on the lung window of chest CT after 2 years of treatment.**

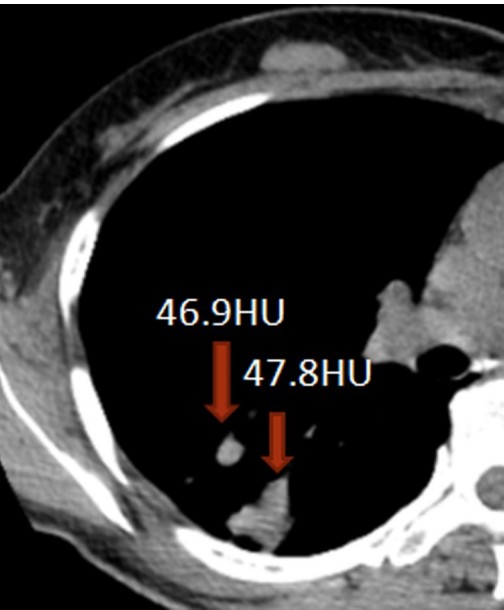

**Fig 15. The volumetric CT value of tuberculomas in the dorsal segment of the right lower lung.**

the sensitivity and specificity of this method were higher than those of conventional B-ultrasound diagnosis. Besson et al. [11] used ITK-SNAP software to segment the FDG PET images of 76 lung tumors confirmed by pathology. At the same time, the ground truth manual segmentation was performed, and the results showed that ITK-SNAP is accurate and reliable for active-contour-based segmentation of heterogeneous thoracic PET tumors and ITK-SNAP surpassed the recommended PET methods compared with ground truth manual segmentation. Qiao et al. [12] selected 93 patients with lung cancer confirmed by pathologic evaluation

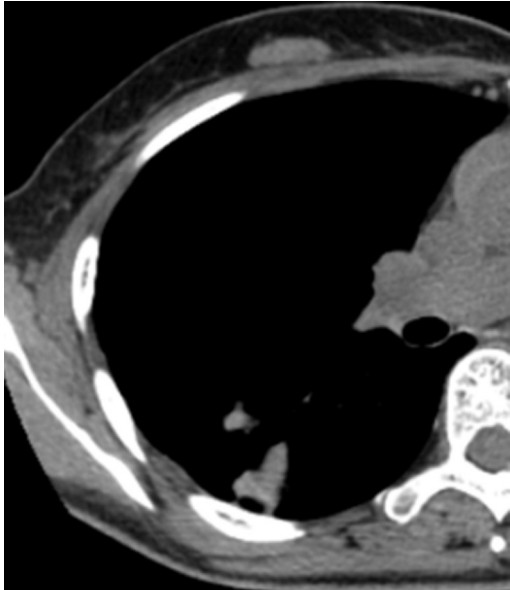

**Fig 16. The tuberculomas on the mediastinal window of chest CT after 1 week of treatment.**

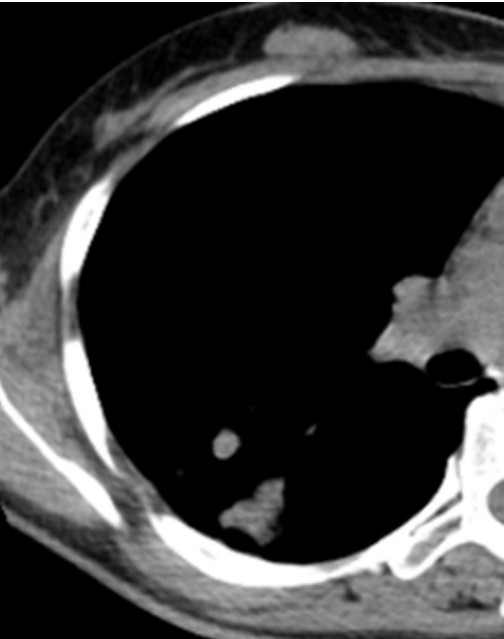

**Fig 17. The tuberculomas on the mediastinal window of chest CT after 1 year of treatment.**

in a study correlating the TP53 gene and imaging characteristics of lung cancer and used ITK-SNAP software to segment lung cancer, then used AK software to extract the radiologic characteristics of lung cancer. The result showed that mutation of TP53 gene was correlated with the imaging characteristics of lung cancer, which has important application value for personalized treatment of lung cancer. In the current study, one of the basic functions of ITK-S-NAP software was to manually segment the tuberculoma focus and obtain the volumetric CT value. In the future, if ITK-SNAP software is combined with radiomics analysis and artificial

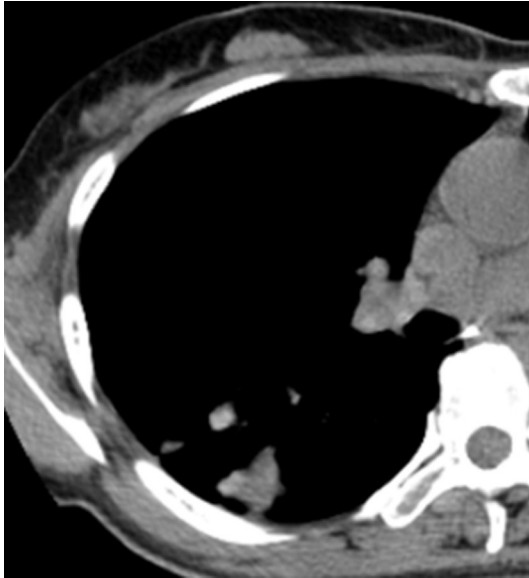

**Fig 18. The tuberculomas on the mediastinal window of chest CT after 2 years of treatment.**

intelligence, it is expected that the results will be more conducive to improving the accuracy of determining pulmonary tuberculosis activity.

Although the volumetric CT value has the advantage of accurately quantifying the activity of pulmonary tuberculomas, the volumetric CT value also has the following limitations: (1) To reduce the impact of the partial volume effect, the research object selected in this study was limited to the lesions with clear boundaries and no exudation on the lung window, and the shortest diameter line on the mediastinal window was >5 mm, while in practical work the proportion of lesions meeting the inclusion criteria is not high, For example, ground glass shadows, the tree bud sign, fine fiber lesions, and millet nodules did not meet the inclusion criteria and there was a selective deviation, which also led to the statistical results of this study being very close to the clinical follow-up results. Therefore, how to make this method more widely applicable to more types of pulmonary tuberculosis lesions remains to be further studied and resolved. (2) Because it takes a long time to manually draw the ROI of each level of tuberculoma focus, the number of samples in this study was limited, and more patients need to be collected to make the results more accurate. (3) The partial volume effect of multi-slice spiral CT will lead to an inaccurate volumetric CT value, especially for smaller lesions. (4) Because this was a retrospective study, the CT scan image did not use the thin layer scanning standard; the scanning conditions will be standardized in a corollary study to obtain more accurate results. All the above factors will lead to deviation in the measurement of the volumetric CT value, which needs to be further improved.

To summarize, the difference between active and inactive pulmonary tuberculomas was quantified by measuring the volumetric CT values of pulmonary tuberculomas. This has important clinical application in judging the activity of pulmonary tuberculomas and it is expected to further improve the accuracy of judging the activity of pulmonary tuberculosis.

## Supporting information

**S1 Data.**
(XLSX)

## Acknowledgments

We thank International Science Editing (http://www.internationalscienceediting.com) for editing this manuscript.

## Author Contributions

**Data curation:** Ganhui Wei, Jiacheng Zhang, Xiaowei Qiu.

**Formal analysis:** Ganhui Wei.

**Investigation:** Ganhui Wei, Jiacheng Zhang.

**Methodology:** Ganhui Wei.

**Project administration:** Ganhui Wei.

**Validation:** Ganhui Wei.

**Writing – original draft:** Ganhui Wei.

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
