## [Decision Letter · Decision Letter 0]

22 Apr 2024

PONE-D-23-15680Application value of volumetric CT value in quantifying the activity of a pulmonary tuberculomaPLOS ONE

Dear Dr. wei,

Thank you for submitting your manuscript to PLOS ONE. After careful consideration, we feel that it has merit but does not fully meet PLOS ONE’s publication criteria as it currently stands. Therefore, we invite you to submit a revised version of the manuscript that addresses the points raised during the review process. The paper presents good evidence of the value of volumetric CT value, but some questions regarding the study must be addressed before publication.

We look forward to receiving your revised manuscript.

Kind regards,

Jose Gerardo Tamez-Peña, PhD

Academic Editor

PLOS ONE

Journal Requirements:

"NO" 

Reviewers' comments:

Reviewer's Responses to Questions

**Comments to the Author**

1. Is the manuscript technically sound, and do the data support the conclusions?

Reviewer #1: Partly

Reviewer #2: Yes

2. Has the statistical analysis been performed appropriately and rigorously? 

Reviewer #1: Yes

Reviewer #2: Yes

3. Have the authors made all data underlying the findings in their manuscript fully available?

Reviewer #1: Yes

Reviewer #2: No

4. Is the manuscript presented in an intelligible fashion and written in standard English?

Reviewer #1: Yes

Reviewer #2: Yes

5. Review Comments to the Author

Reviewer #1: It is a retrospective study. CT study was used were approx 5mm slice thickness and it usually not give good 3-D reconstruction. How the authors assess the cavitary and fibrous inactive lesions ? Do all patients treatment follow up were available? Drug resistant cases also common, so do the authors included those cases ? Pretreatment and post treatment cases were analyzed in the study and how ?

Partial volume effect in the 3-D images and AI also an issue with these techniques.

Reviewer #2: I'm pleased to have a precious opportunity to review the interesting paper on Application value of volumetric CT value in quantifying the activity of a pulmonary

Tuberculoma

Major Strengths:

1. Good concept and clinically useful study

2. Simple and lucid methodology

Major Weaknesses:

1. Retrospective single centre study

Title:

* Ok

Abstract:

* Generally ok

• Page 2 ,line 17: “shortest diameter line”; change to “shortest diameter axis”

Key Words:

* Add Pulmonary Tuberculosis

Introduction:

* Generally OK.

Methods

* Authors should specify where do their patients come from (ED/DH/general

ward...)

Page 6, line 100: Kindly explain : “significantly enlarged or decreased in size”

* No other relevant flaws about methods.

Results:

* Generally ok.

Discussion:

* ok

Conclusion:

* Generally ok.

Figures

Good images

6. PLOS authors have the option to publish the peer review history of their article (what does this mean?). If published, this will include your full peer review and any attached files.

Reviewer #1: **Yes: **Dr.Pratiksha Yadav

Reviewer #2: No

---

## [Author Response · Author response to Decision Letter 0]

24 May 2024

Reviewer #1: It is a retrospective study. CT study was used were approx 5mm slice thickness and it usually not give good 3-D reconstruction. How the authors assess the cavitary and fibrous inactive lesions ? Do all patients treatment follow up were available? Drug resistant cases also common, so do the authors included those cases ? Pretreatment and post treatment cases were analyzed in the study and how ?

Because this was a retrospective study, thin-layer scanning technology was not used for the CT scan images. The reconstructed layer thickness of the CT mediastinal window was 5mm. The scanning layer thickness and interval were 2.5 mm. The reconstruction layer thickness of the mediastinal window was 5 mm and the matrix was 512 × 512.The research method in this study involved importing CT images from pulmonary tuberculoma patients into ITK-SNAP software using DICOM format, locating the tuberculoma in the ITK-SNAP software, and delineating the ROI of the tuberculoma at each level. Then, the volume and statistical options of the software were used to calculate the volumetric CT value. All operations are performed on the CT cross-section without 3D reconstruction.

（1）How the authors assess the cavitary and fibrous inactive lesions ?

A：How the authors assess the cavitary lesions ?

The research object chosen by the author in this study was limited to tuberculomas with clear boundaries and no exudative lesions in the lung window, and tuberculomas with a shortest diameter >5mm in the mediastinal window.

Of tuberculomas with a cavity, only tuberculomas with a thick-wall cavity and short wall diameter>5mm were included in the study. If the cavity wall did not exceed 5mm, the tuberculoma was not included in the study. The images of pulmonary tuberculomas with thick-walled cavities were included in ITK-SNAP software using the DICOM format.The tuberculoma was found using the ITK-SNAP software and the cavity walls on each level were delineated. To avoid errors, the ROI did not include normal lung tissue, cavities, or gas components that did not contain pathologic tissues.The volumetric CT value of the lesion was calculated using the volume and statistical options in the software. Finally, 45.32HU was used as the threshold to determine the activity of the tuberculoma cavity. If the volumetric CT value was > 45.32HU, the tuberculoma cavity was considered inactive. If the volumetric CT value was < 45.32HU, the tuberculoma cavity was considered active.

B：How the authors assess the fibrous inactive lesions ?

The research object selected by the author in this study was limited to tuberculomas with clear boundaries and no exudative lesions in the lung window and lesions with a shortest diameter > 5mm on the mediastinal window. Therefore, fibrous lesions were not included.

（2）Do all patients treatment follow up were available?

All patients in this study had follow-up evaluations.Patients clinically diagnosed with active pulmonary tuberculoma underwent chest CT re-examinations during anti-tuberculosis treatment. Patients clinically diagnosed with inactive pulmonary tuberculoma did not receive anti-tuberculosis treatment and continued to undergo regular chest CT scans.

（3）Drug resistant cases also common, so do the authors included those cases ?

The scope of this study was tuberculomas with clear boundaries, non-exudative lesions on the lung window and a shortest diameter > 5mm on the mediastinal window. Therefore, the scope of this study also included drug-resistant tuberculoma cases. After formal anti-tuberculosis treatment, drug-resistant tuberculoma still had activity, and the volumetric CT value did not significantly increase or decrease (always <45.32HU). The drug-resistant tuberculoma was judged as an active tuberculoma and , which was consistent with the clinical diagnosis.Therefore, in this study the sensitivity, specificity, and accuracy using volumetric CT values determined the activity of pulmonary tuberculomas and were not affected by drug resistance.

（5）Pretreatment and post treatment cases were analyzed in the study and how ?

In the process of anti-tuberculosis treatment, except for a few tuberculomas that were completely absorbed after treatment, the majority of active tuberculomas transformed from a soft tuberculoma focus to a hard tuberculoma focus [9]. Tuberculomas are absorbed during treatment, the volume decreases, and the density increases [9]. When the density and volumetric CT values reach a specific threshold, the focus is in a stable state, the lesion size will not increase or decrease, the symptoms will resolve, the bacteriologic examination becomes negative, and the active tuberculoma will become an inactive tuberculoma [9].

[9]Lee HS, Oh JY, Lee JH, Yoo CG, Lee CT, Kim YW, et al. Response of pulmonary tuberculomas to anti-tuberculous treatment. EurRespir J. 2004 Mar; 23(3): 452–455. doi: 10.1183/09031936.04.00087304. PMID: 15065838.

Lee et al. [6] made dynamic observations on high-resolution CT films of 52 sputum-positive pulmonary tuberculosis patients from before or after <2 weeks of anti-tuberculosis treatment to 6–9 months of anti-tuberculosis treatment and reported that tuberculosis foci were loose and low density before anti-tuberculosis treatment. After anti-tuberculosis treatment, with an extension of treatment time, the tuberculosis focus shrinks, the density increases, and the calcification focus often increases.

[6] Lee JJ, Chong PY, Lin CB, Hsu AH, Lee CC. High resolution chest CT in patients with pulmonary tuberculosis: characteristic findings before and after antituberculous therapy. Eur J Radiol. 2008 Jul; 67(1): 100–104. doi: 10.1016/j.ejrad.2007.07.009. Epub 2007 Sep 17. PMID: 17870275.

In this study we also found that the volumetric CT value of active pulmonary tuberculoma was lower before treatment (<45.32HU),while after formal anti-tuberculosis treatment the volumetric CT value of the tuberculoma gradually increased. When the volumetric CT value was greater than the threshold (45.32HU), the lesion would stabilize and not enlarge or shrink, thus becoming inactive pulmonary tuberculoma.

Therefore, specific analysis of the tuberculoma activity before and after treatment is as follows.For the initial diagnosis of a tuberculoma, chest CT is performed on the patient, and the volumetric CT value of the tuberculomas measured. If the volumetric CT value is >45.32HU, the tuberculomas judged as non-active and does not require treatment.If the volumetric CT value is <45.32, the tuberculoma is judged as active and anti-tuberculosis treatment is required. During the treatment process, the chest CT should be regularly reviewed to measure the volumetric CT value of the tuberculoma. After formal and effective anti-tuberculosis treatment, the volumetric CT value of the tuberculoma will gradually increase and the volume will gradually decrease. When the volumetric CT value is>45.32HU, the tuberculoma lesion is considered stable and treatment can be discontinued.

Therefore, the threshold of the volume CT values can accurately determine the activity of pulmonary tuberculomas before and after treatment, which is conducive to providing patients with accurate treatment. Tuberculoma recurrence will not occur due to insufficient treatment nor will harm come to patients due to drug side effects from excessive treatment.

Partial volume effect in the 3-D images and AI also an issue with these techniques.

Yes, both manual and artificial intelligence measurements of volumetric CT values will have a partial volume effect. The method adopted in this study is as follows. When outlining the ROI, the lesion is delineated on the CT mediastinal window and retreated 1-2 pixels in each direction of the lesion boundary to reduce the influence of partial volume effects.

Reviewer #2: I'm pleased to have a precious opportunity to review the interesting paper on Application value of volumetric CT value in quantifying the activity of a pulmonary

Tuberculoma

Major Strengths:

1. Good concept and clinically useful study

2. Simple and lucid methodology

Major Weaknesses:

1. Retrospective single centre study

 Hangzhou Red Cross Hospital is a designated hospital for pulmonary tuberculosis in Hangzhou, Zhejiang Province. Hangzhou Red Cross Hospital has a large tuberculosis patient population, a complete range of types, and complete equipment. Other hospitals in Hangzhou send tuberculosis patients to the hospital for treatment. Therefore, there are very few tuberculosis patients and CT images in other Hangzhou hospitals. As a result, the author only chose the CT images of tuberculomas from Hangzhou Red Cross Hospital for research purposes.

Title:

* Ok

Abstract:

* Generally ok

• Page 2 ,line 17: “shortest diameter line”; change to “shortest diameter axis”

Revised in the paper.

Key Words:

* Add Pulmonary Tuberculosis

Added in the paper

Introduction:

* Generally OK.

Methods

* Authors should specify where do their patients come from (ED/DH/general

ward...)

All patients were enrolled from the Tuberculosis Outpatient or Inpatient Departments of Hangzhou Red Cross Hospital

Page 6, line 100: Kindly explain : “significantly enlarged or decreased in size”

Considering the accuracy of the description and referring to previous literature, it is more appropriate to remove the word “significantly”.

The criteria for judging the activity of tuberculomas were as follows: after 2 years of follow-up evaluations, the patients in the group were judged to have an inactive pulmonary tuberculoma if the tuberculoma focus was stable, no clinical symptoms and signs related to active pulmonary tuberculosis, and the bacteriologic examination was negative[5].if the stability of the tuberculoma cannot be established and enlarged or decreased in size (except for the possibility of secondary diseases), the focus of the tuberculoma was judged to be an active pulmonary tuberculoma whether or not there were clinical symptoms and signs related to active pulmonary tuberculosis and whether or not the bacteriologic examination was positive or negative[5].

[5]Nachiappan AC, Rahbar K, Shi X, Guy ES, Mortani Barbosa EJ Jr, Shroff GS, Ocazionez D, Schlesinger AE, Katz SI, Hammer MM. Pulmonary Tuberculosis: Role of Radiology in Diagnosis and Management. Radiographics. 2017 Jan-Feb;37(1):52-72.

Enlarged or decreased volume of tuberculoma indicates that the tuberculosis lesion is unstable and active.

Enlargement of the lesion indicates that the initially diagnosed tuberculoma is unstable and active, so the lesion will progress over time, usually referring to patients with active tuberculoma who have not received treatment or formal treatment or after formal treatment, but the tuberculoma has drug resistance and disease progression with increased volume; these tuberculomas are all active tuberculomas.

The reduction of the tuberculoma lesion indicates that the original tuberculoma is also unstable and active. Tuberculoma lesion reduction usually refers to absorption, volume reduction, and increased density of the tuberculoma after formal and effective anti-tuberculosis treatment, gradually transforming from active tuberculoma to inactive pulmonary tuberculoma. Therefore, reduction of the tuberculoma lesion during re-examination can prove that the initially diagnosed tuberculoma is active.

6. PLOS authors have the option to publish the peer review history of their article. If published, this will include your full peer review and any attached files.

Yes

Do you want your identity to be public for this peer review? For information about this choice, including consent withdrawal, please see our Privacy Policy.

Reviewer #1: Yes: Dr.Pratiksha Yadav

Reviewer #2: No

Yes

---

## [Decision Letter · Decision Letter 1]

25 Jun 2024

Application value of volumetric CT value in quantifying the activity of a pulmonary tuberculoma

PONE-D-23-15680R1

Dear Dr. wei,

We’re pleased to inform you that your manuscript has been judged scientifically suitable for publication and will be formally accepted for publication once it meets all outstanding technical requirements.

Kind regards,

Jose Gerardo Tamez-Peña, PhD

Academic Editor

PLOS ONE

Additional Editor Comments (optional):

Reviewers' comments:

Reviewer's Responses to Questions

**Comments to the Author**

1. If the authors have adequately addressed your comments raised in a previous round of review and you feel that this manuscript is now acceptable for publication, you may indicate that here to bypass the “Comments to the Author” section, enter your conflict of interest statement in the “Confidential to Editor” section, and submit your "Accept" recommendation.

Reviewer #1: All comments have been addressed

Reviewer #2: All comments have been addressed

2. Is the manuscript technically sound, and do the data support the conclusions?

Reviewer #1: Yes

Reviewer #2: Yes

3. Has the statistical analysis been performed appropriately and rigorously? 

Reviewer #1: Yes

Reviewer #2: Yes

4. Have the authors made all data underlying the findings in their manuscript fully available?

Reviewer #1: Yes

Reviewer #2: No

5. Is the manuscript presented in an intelligible fashion and written in standard English?

Reviewer #1: Yes

Reviewer #2: Yes

6. Review Comments to the Author

Reviewer #1: Authors revised manuscript and covers the points. Use arrows in the images to demonstrate the abnormality/lesion.

Reviewer #2: Good efforts. All comments in the previous round of review have been addressed. The manuscript can be accepted

7. PLOS authors have the option to publish the peer review history of their article (what does this mean?). If published, this will include your full peer review and any attached files.

Reviewer #1: No

Reviewer #2: No

---

## [Editor Report · Acceptance letter]

2 Aug 2024

PONE-D-23-15680R1 

PLOS ONE

Dear Dr. wei, 

I'm pleased to inform you that your manuscript has been deemed suitable for publication in PLOS ONE. Congratulations! Your manuscript is now being handed over to our production team.

Kind regards, 

on behalf of

Dr. Jose Gerardo Tamez-Peña 

Academic Editor

PLOS ONE